# Adaptive Dataset Management Scheme for Lightweight Federated Learning in Mobile Edge Computing

**DOI:** 10.3390/s24082579

**Published:** 2024-04-18

**Authors:** Jingyeom Kim, Juneseok Bang, Joohyung Lee

**Affiliations:** School of Computing, Gachon University, Seongnam 13120, Republic of Korea; kimo1113@gachon.ac.kr (J.K.); dipsy1234@gachon.ac.kr (J.B.)

**Keywords:** federated learning, mobile edge computing, dataset management

## Abstract

Federated learning (FL) in mobile edge computing has emerged as a promising machine-learning paradigm in the Internet of Things, enabling distributed training without exposing private data. It allows multiple mobile devices (MDs) to collaboratively create a global model. FL not only addresses the issue of private data exposure but also alleviates the burden on a centralized server, which is common in conventional centralized learning. However, a critical issue in FL is the imposed computing for local training on multiple MDs, which often have limited computing capabilities. This limitation poses a challenge for MDs to actively contribute to the training process. To tackle this problem, this paper proposes an adaptive dataset management (ADM) scheme, aiming to reduce the burden of local training on MDs. Through an empirical study on the influence of dataset size on accuracy improvement over communication rounds, we confirm that the amount of dataset has a reduced impact on accuracy gain. Based on this finding, we introduce a discount factor that represents the reduced impact of the size of the dataset on the accuracy gain over communication rounds. To address the ADM problem, which involves determining how much the dataset should be reduced over classes while considering both the proposed discounting factor and Kullback–Leibler divergence (KLD), a theoretical framework is presented. The ADM problem is a non-convex optimization problem. To solve it, we propose a greedy-based heuristic algorithm that determines a suboptimal solution with low complexity. Simulation results demonstrate that our proposed scheme effectively alleviates the training burden on MDs while maintaining acceptable training accuracy.

## 1. Introduction

With the success of deep learning (DL) and the rich storage and computing capabilities of mobile devices (MDs), many applications utilizing DL with the collected data from MDs, such as face recognition, recommendation systems, and human activity recognition, have become widely employed. Traditionally, the centralized learning method has been used for training DL models, which requires collecting all raw data from MDs by sending raw data to a remote cloud server. However, this approach raises concerns about MDs’ privacy and potential data misuse [1]. To address these challenges, a promising distributed learning framework called federated learning (FL) has emerged [2,3]. FL allows for the construction of global DL models using only local model weights trained from MDs with their local datasets. This approach helps alleviate privacy concerns and distributes the training burden across multiple MDs at a centralized server. Nevertheless, efficient management of FL still faces several challenging issues due to its distributed and heterogeneous nature. Specifically, FL suffers from long transmission latency to a remote cloud server [4]. Additionally, the varying qualities of each MD’s dataset can impact the accuracy of the global model, and the local training burden may make MDs reluctant to participate in the FL process [5,6,7].

To tackle these challenges, researchers have explored the concept of mobile edge computing (MEC)-assisted FL, which leverages the computing capabilities available at the mobile network edge to facilitate intermediate aggregation and streamline the FL process effectively [8]. The objective is to achieve optimal FL operations. In particular, there have been active investigations into efficient FL participant selection, aggregation management, and radio and computation resource management (such as CPU/GPU). In terms of radio and computation resource management, given the constraints of limited communication bandwidth and computational capacity, various approaches to joint edge association and the allocation of radio and computation resources have been explored. These endeavors aim to enhance the accuracy of the global model or improve the speed of convergence as well as energy efficiency [6,9,10,11,12,13,14,15,16,17,18,19].

However, most existing research assumes that participating MDs fully utilize their given datasets, considering the fact that more data generally leads to higher accuracy in DL. In practice, however, MDs may have limited computing and battery capabilities, making it burdensome for MDs if they have large datasets. The burden placed on MDs due to the full utilization of their datasets in the FL process often leads to a reduction in their participation, resulting in a decrease in global accuracy. Thus, this issue requires careful management. Nevertheless, current studies tend to focus on providing incentives to promote MD participation rather than addressing the burden on MDs [20,21,22]. Only a few studies have considered dataset management for efficient data utilization. For instance, Duan et al. [7] proposed the Astrea framework, which adaptively conducts data augmentation and downsampling to address local data imbalance. Similarly, Ren et al. [9] proposed a joint batch size selection and communication resource allocation algorithm for both CPU and GPU resources, taking into account the trade-off between loss decay and latency. Additionally, Kim et al. [6] proposed an energy-efficient joint dataset and computation management scheme that utilizes dataset management to reduce the burden on MDs, enabling energy-efficient FL with minimal accuracy loss. However, most of these works assume that the value of the dataset remains constant throughout the FL process. In non-independent and identically distributed (non-IID) cases, both the amount of data and the class distribution across rounds can impact the global accuracy as mentioned in [23,24]. To the best of our knowledge, the careful consideration of dataset management for alleviating the burden on MEC-assisted FL processes with minimal accuracy loss, specifically addressing the diminishing effect of dataset size over rounds and class distribution, has not been thoroughly explored in previous studies. This constraint highlights an area that requires further attention and research.

In this article, we propose an adaptive dataset management (ADM) scheme to alleviate the burden on MDs in the FL process while minimizing accuracy loss. The detailed contributions of this article are as follows:Considering the diminishing effect of dataset size over rounds, as validated by our empirical study, along with the distribution of classes, we design the proposed ADM scheme. This scheme incorporates both the data size adjustment algorithm and the class adjustment algorithm.To develop the proposed ADM scheme, we present rigorous analytical models to estimate accuracy and end-to-end service latency concerning the dataset size, taking into account the proposed discount factor. Subsequently, to balance between the estimated accuracy and end-to-end service latency, we formulate the objective function based on the ratio of these two factors.Regarding the dataset size adjustment problem, we determine the optimal dataset size adjustment across clients by solving an optimization problem, initially a non-convex problem due to the presence of a non-differentiable function. We employ several mathematical techniques to transform it into a convex optimization problem and provide the global optimum solution.In addressing the class adjustment problem, we establish the optimal dataset size adjustment across classes for each client, considering the class distribution over MDs in the non-IID case. Here, we propose a greedy-based heuristic algorithm to reduce the Kullback–Leibler Divergence (KLD) distance and derive a suboptimal solution with low complexity.As a practical consideration, we also provide a detailed discussion on the implementation of the proposed ADM on a virtualization platform, along with a prototype of the proposed framework. Finally, simulation results demonstrate the effectiveness of the proposed scheme in reducing the training burden on MDs while maintaining acceptable training accuracy.

The remainder of this paper is organized as follows. Section 2 summarizes the previous works related to resource management for MEC-assisted FL. Section 3 provides the proposed system model of the ADM scheme. Section 4 provides our proposed ADM scheme, which consists of (i) dataset size adjustment and (ii) class adjustment and detailed discussions on its implementation. In Section 5 evaluates the performance of the proposed system. Finally, Section 6 concludes the paper.

## 2. Related Work

Research on MEC-assisted FL has seen numerous approaches in the literature aimed at achieving optimal dataset management of FL. It is important to note that effective data management is essential for achieving optimal performance in Internet of Things (IoT) systems. In particular, as FL has been extensively utilized in IoT systems, there is a growing interest in addressing this issue to enhance FL performance [25]. In this brief survey, we will specifically focus on recent advancements in dataset management for MEC-assisted FL. The work by [9] focused on accelerating the training process in CPU and GPU computation scenarios. They proposed a joint batch size selection and radio resource allocation algorithm. This approach aimed to optimize the training process by effectively utilizing both CPU and GPU resources. In [6], the authors developed an analytical model to optimize learning efficiency and energy consumption. They proposed a joint dataset and computation management approach to strike a balance between these two factors. In [26], the authors proposed an adaptive batch size and learning rate selection algorithm. Their objective was to mitigate the negative impact of the synchronization barrier in FL. The authors of [27] proposed a detailed data selection method to enhance learning efficiency by choosing only the data that contributes to the improvement of the model’s performance. However, these studies did not take into account the distribution of data. Other research has focused on considering the distribution of data, which can be classified into data adjustment and client selection/scheduling approaches. In [7], the authors proposed a self-balancing algorithm to alleviate data imbalance. This was achieved by augmenting the minority class and downsampling the majority class on MDs. Additionally, they proposed a mediator-based rescheduling algorithm to select MDs and distribute aggregated data to achieve near-IID distribution. In [28], a data augmentation strategy using a generative adversarial network (GAN) was employed. This strategy enabled each device to locally reproduce the data samples of all MDs, therefore promoting an IID distribution of data. The authors of [29] introduced FedSwap, a method that swaps models among the MDs in each round to alleviate data diversity. This approach aimed to achieve a more consistent data distribution across devices. In [30], the authors proposed the CSFedAvg algorithm, which selects clients with a low degree of non-IID data by utilizing weight divergence. Similarly, The authors of [31] employed DRL to identify clients relevant to the application task, ensuring they have uniform data across a large dataset. These strategies are designed to enhance the aggregation process by focusing on clients with similar data distributions. However, these approaches, which focus on aligning IID data through data augmentation or client selection, tend to overlook the learning efficiency of MDs. We propose ADM, taking into consideration the aforementioned limitations. ADM accelerates the learning process by adjusting the dataset in response to diminishing the effect of dataset size over rounds while also adjusting the class distribution by considering non-IID data of MDs. We present a list summary of related works in Table 1.

## 3. System Model

### 3.1. Motivating Example for Discounting Factor

It is well-established that the training accuracy of FL generally improves as the dataset size per MD increases. However, the rate at which the accuracy increment, also known as accuracy gain, tends to decrease as the communication rounds progress [6,18]. To further investigate the impact of dataset size on accuracy improvement over communication rounds, we conducted an empirical study, which led us to introduce a discount factor in our proposed scheme.

Our simulations, as depicted in Figure 1 and Figure 2, demonstrate the accuracy trend of the global model in the FL framework when using the CIFAR-10 and MNIST datasets. Under both the Independent and Identically Distributed (IID) and non-IID settings, we observed that the accuracy increases with an increasing amount of dataset denoted as $D_{n}$, where *n* represents the index of MDs at a specific communication round, as confirmed by previous studies [6,18]. However, in this experiment, where the total number of MDs is 10, we noted that the accuracy improvement from utilizing a larger dataset size, $D_{n}$, diminishes as the communication rounds progress. Notably, the gap between $D_{n}$ = 1000 and $D_{n}$ = 2500 decreases consistently, supporting this observation. This trend becomes more evident when we analyze the accuracy gain based on communication rounds with respect to $D_{n}$, as illustrated in Figure 3 and Figure 4. These demonstrate that a significant accuracy gain occurs in the initial rounds, but after 40 rounds, the accuracy gain significantly decreases. The high variance observed in the initial rounds reflects greater volatility in accuracy gain compared to other rounds. This indicates a tendency for accuracy to increase significantly in the early rounds. Based on these findings, we can conclude that to alleviate the training burden on MDs, it is advisable to reduce the dataset size more aggressively as the communication rounds evolve. To account for this trend in adaptive data management, we propose the introduction of a discounting factor that considers the accuracy gained from the dataset in relation to the communication rounds.

### 3.2. Proposed System Model

For ease of reference, we present Table 2, which comprises a list of the key symbols that we define and utilize in this paper.

As depicted in Figure 5, the proposed system architecture supports an FL framework consisting of multiple MDs and a single MEC server where the MEC server is directly connected to a single base station (BS) serving MDs. Here, we define **N** as the set of MDs, where |N| = *N* denotes the total number of MDs. For each n∈N, MD *n* has local dataset $D_{n}=(\left(x_{n}^{1},y_{n}^{1}\right),\left(x_{n}^{2},y_{n}^{2}\right),\left(x_{n}^{3},y_{n}^{3}\right),$ …, $(x_{n}^{i},y_{n}^{i}),$ …, $\left(x_{n}^{|D_{n}|},y_{n}^{|D_{n}|})\right)$, where $|D_{n}|$ = $D_{n}$ denotes the total number of samples, $x_{n}^{i}$ and $y_{n}^{i}$ is the *i*-th data sample and corresponding ground-truth label, respectively.

In this context, the MEC server, acting as a centralized server, orchestrates the coordination of FL tasks across multiple MDs. Each MD performs local training on its local dataset and sends the local update to the MEC server. Subsequently, the MEC server aggregates the local updates from the MDs to generate the global model. Throughout this process, the MEC server employs an adaptive data management (ADM) scheme to adjust the dataset size assigned to each MD, therefore achieving the lightweight FL process. The entire procedure can be categorized into the following three steps:Step 1: In Step 1, the MEC server selects appropriate MDs as FL participants. Then, the MEC server requests and receives the class distribution of the dataset from each MD to conduct the ADM scheme, which will be explained in detail in Section IV. Using the information obtained from the selected MDs, the data adjustment message is calculated using the ADM scheme. Afterward, the MEC server initiates the task by providing an initial shared global model, denoted as $w_{G}^{0}$, and the data adjustment message for local training to multiple MDs. The initial shared global model may include a TensorFlow graph, weights, and instructions.Step 2: Each MD *n* conducts local training on the adjusted dataset $D_{n}^{\prime }$ among entire local data $D_{n}$ using the shared global model (i.e., $w_{G}^{0}$ in the initial round or $w_{G}^{t}$ in round *t*). Specifically, by minimizing the loss function $L(w_{n})$, the local model parameter $w_{n}$ at MD *n* is given by
(1)$$w_{n}^{*}=\arg\min_{w_{n}}L\left(w_{n}\right).$$Then, the updates are transferred to the MEC server.Step 3: The MEC server combines the local model updates from the MDs and generates a global model by solving an optimization problem that minimizes the global loss function.
(2)$$L\left(w_{G}^{t}\right)=\frac{1}{N}\sum_{i\in I}L\left(w_{i}\right).$$Then, the MEC server sends the updated global model parameters back to the MDs.

Multiple rounds of the FL process, which include Steps 2–3, are iterated until either the global loss function at the MEC server converges to the termination condition or the specified target accuracy is achieved. It should be noted that the FL process allows for the selection of different types of ML models depending on the specific application of the FL service. Additionally, in Step 3, the aggregation of the global model, an essential component of FL, can be accomplished using various mechanisms such as the federated averaging algorithm and secure aggregation algorithm [32].

### 3.3. Analytical Models

In this subsection, we represent analytical models by formulating (i) an accuracy estimation model and (ii) an end-to-end service latency model, respectively.

#### 3.3.1. Accuracy Estimation Model

In most of the literature [6,18], the estimated training accuracy in FL can be modeled as either a concave or linear function, depending on the observation range. The concave behavior of the accuracy can be approximated by piecewise linear approximation. To simplify our proposed scheme and control the dataset size within specific ranges, as shown in [6], we adopt a linear function in this paper. Furthermore, considering the impact of dataset size on accuracy improvement over communication rounds, we introduce a discounting factor (a number between 0–1), which provides a clever way to scale down the impact of accuracy improvement increasingly after each round evolves. Then, the proposed accuracy estimation model *A* is given by
(3)$$A=\sigma^{r-1}\sum_{n\in N}D_{n},$$
where σ is the discounting factor and *r* is a communication round index. It represents the extent to which the accuracy improvement is discounted with respect to the dataset size of MDs over communication rounds. If $\sigma_{n}=1$, the accuracy improvement remains sustained regardless of the communication rounds. However, when $\sigma_{n}<1$, the impact of the dataset size on accuracy improvement gradually diminishes as the number of communication rounds increases.

#### 3.3.2. End-to-End Service Latency Model

Following [6], the end-to-end service latency comprises two components: (i) computation latency and (ii) transmission latency between MD *n* and the MEC server. The computation latency includes the MD *n*’s local model computation latency, denoted as $L_{n}^{c}$, and the MEC server’s computation latency for global model aggregation. Considering that global model aggregation is typically a lighter task compared to local model training, we assume that the latency associated with global model aggregation is negligible. This assumption aligns with various studies [6] that assume the MEC server possesses sufficient resources to handle this task. Let $c_{n}$ be the number of CPU cycles of MD *n* required to process one sample, and $f_{n}$ is the CPU frequency. Then, the local model computation latency, $L_{n}^{c}$, can be expressed as
(4)$$L_{n}^{c}=I_{n}\frac{c_{n}D_{n}}{f_{n}},$$
where the $I_{n}$ denotes the number of training local model iterations.

The transmission latency, denoted as $L_{n}^{t}$, encompasses both the uploading of the local model and the downloading of the global model. However, since the latency associated with downloading the global model is negligible compared to uploading the local model, we can disregard it in our analysis, as assumed in [6]. To formulate the transmission latency $L_{n}^{t}$, we consider that the MEC server fairly assigns bandwidth to all MDs, with an available bandwidth denoted as *B*. The transmission rate of MD *n* can be defined as:(5)$$R_{n}=\frac{B}{N}\ln\left(1+\frac{h_{n}p_{n}}{N_{0}}\right),$$
where $h_{n}$, $p_{n}$, and $N_{0}$ represent the channel gain, transmission power of MD *n*, and the background noise, respectively. The transmission latency for the local model parameters $w_{n}$ from MD *n* to the MEC server can be formulated as:(6)$$L_{n}^{t}=\frac{W_{n}}{R_{n}},$$
where $W_{n}$ denotes the size of the local model parameters of MD *n*.

Finally, the end-to-end service latency of each MD *n* is defined as $L_{n}=L_{n}^{c}+L_{n}^{t}$. In FL, the global model aggregation is conducted when the MEC server receives all local models from all MDs, so the total end-to-end service latency of FL, which is determined by the slowest MD, is defined as
(7)$$L=\max_{n\in N}\left\{L_{n}^{c}+L_{n}^{t}\right\}.$$

## 4. Proposed ADM Scheme

In this section, we present a novel adaptive dataset management (ADM) scheme for lightweight FL frameworks. The goal of this scheme is to determine the optimal dataset adjustment vector $v_{n}$, which represents the ratio between the amount of local dataset used for local training and the total amount of local dataset. Additionally, we address the problem of determining which class of data should be reduced, taking into account the class distribution.

### 4.1. Dataset Size Adjustment

To strike a balance between accuracy improvement and end-to-end service latency with respect to the dataset size, we define an optimization problem aimed at maximizing the ratio of accuracy estimation *A* to end-to-end service latency *L* in FL. In this formulation, we replace $D_{n}$ with the adjusted dataset $D_{n}^{\prime }$, where $D_{n}^{\prime }=v_{n}D_{n}$. Then, the problem can be formulated as follows:

*Prob*.1:(8a)maxvn  AL(8b)s.t.   γd≤vn≤1,∀n∈N,
where the constraint (8b) specifies the range of $v_{n}$ with lower bound parameter $\gamma_{d}$ for the selected MEC MDs.

To convert Prob.1 into standard minimization form by plugging (Equation 3)–(Equation 7) into Prob.1, the Prob.1 is newly defined as follows:

Prob.2:(9a)minvn  −σr−1∑n∈NvnDnmaxn∈N{IncnvnDnfn+WnRn}(9b)s.t.   γd≤vn≤1,∀n∈N,

However, since Prob.2 has the form max(.) in the objective function, which is not differentiable, it is a non-convex optimization problem. Thus, we convert max(.) into an affine function by introducing an additional variable *t* and letting
(10)$$t=\max_{n\in N}I_{n}\frac{c_{n}v_{n}D_{n}}{f_{n}}+\frac{W_{n}}{R_{n}}.$$

This new variable *t* induces an additional constraint: (11)$$I_{n}\frac{c_{n}v_{n}D_{n}}{f_{n}}+\frac{W_{n}}{r_{n}}\leq t,\forall n\in N.$$

Using this additional constraint with the new variable *t*, we can rewrite Prob.2, which is a problem equivalent to

Prob.3:(12a)minvn,t  −σr−1∑n∈NvnDnt(12b)s.t.    γd≤vn≤1,∀n∈N(12c)          IncnvnDnfn+WnRn≤t,∀n∈N.

**Lemma** **1.**
*Prob. 3 is a linear programming (LP) problem with respect to optimization variables ($v_{n}$).*


**Proof.** First, the objective function is linear with respect to $v_{n}$. Moreover, the inequality constraints (12b) and (12c) are affine in terms of the optimization variables ($v_{n}$). Consequently, since both the objective function and the inequality constraints are affine, the problem is an LP problem in terms of the optimization variables ($v_{n}$).    □

**Lemma** **2.**
*The Prob. 3 is a strictly increasing function with respect to optimization variable (t), and $t^{*}$ should be at the lower bound of constraint (12c).*


**Proof.** As Prob.3 includes the term $-\frac{1}{t}$ in the objective function, it is evident that the objective function is strictly increasing with respect to *t*. Consequently, the optimal value of *t* ($t^{*}$) resides at the lower bound of (12c).    □

Lemmas 1 and 2 form the basis for solving Prob.3 using the block coordinate descent method [33]. In this method, given a fixed value of *t*, we can readily solve for the optimal values of $v_{n}$ using the Simplex algorithm (SA). Subsequently, $t^{*}$ and ($v_{n}$) are obtained iteratively by mutually fixing each other until the cost function converges, following the block coordinate descent approach. The algorithmic procedure is summarized in Algorithm 1.

**Lemma** **3.**
*Algorithm 1 performs a sublinear convergence rate.*


**Proof.** The block coordinate descent method demonstrates a sublinear convergence rate, as proved in [34]. Based on Lemmas 1 and 2, the proposed algorithm is designed using the block coordinate descent method. Consequently, Algorithm 1 has a sublinear convergence rate.    □

**Algorithm 1** ADM scheme—data size adjustment**Input:** Dn,r,σ,Wn,cn,fn,In,Rn,θc**Initialize:** vn and *t* are randomly initialized within the constraints.
**Output:** Optimal vn*Dn=Dn′*
 1: **while** True **do**
 2:  $v_{n}\leftarrow$ solving **Prob.3** via SA
 3:  t←$\max_{n\in N}I_{n}\frac{c_{n}v_{n}D_{n}}{f_{n}}+\frac{W_{n}}{R_{n}}$
 4:  $C_{i}\leftarrow$(12a)
 5:  **if** $|C_{i}-C_{i-1}|<\theta_{c}$ **then**
 6:    **break**
 7:  **end if**
 8:  i = i + 1
 9: **end while**
 10:  $v_{n}^{*}\leftarrow v_{n}$
 11:  $D_{n}^{\prime *}\leftarrow \left\lfloor v_{n}^{*}D_{n}\right\rfloor$


### 4.2. Class Adjustment

After obtaining $D_{n}^{\prime }$ from the previous Algorithm 1, under the assumption of an IID case, it is obvious that all MDs can reduce their entire class of dataset evenly to achieve $D_{n}^{\prime *}$ which is smaller than $D_{n}$. However, in non-IID cases, we should carefully determine which class of data should be reduced, taking into account the class distribution. Specifically, during data reduction, the proposed ADM aims to achieve a uniform distribution of a class of entire datasets across MDs, known as an IID. This approach minimizes the accuracy loss resulting from data adjustment while reducing the training burden on MDs. Here, we define **K** as the set of classes in the dataset, where |K| = *K* denotes the total number of classes. To accomplish this, we utilize the Kullback–Leibler divergence (KLD) distance to quantify the proximity between the class distribution of the dataset and the uniform distribution, which is formulated by
(13)$$D_{KL}(P_{N}\left|\right|P_{u})=\sum_{k\in K}P_{N}\left(k\right)\ln\left(\frac{P_{N}\left(k\right)}{P_{u}}\right),$$
where the $P_{u}$ is the uniform distribution and $P_{N}$ is the class distribution of the entire dataset from all MDs.

For each n∈N, consider that local dataset of MD *n*
$D_{n}$ has disjoint subset $D_{n,k}$, which satisfies
(14)$$D_{n}=\underset{k\in K}{⋃}D_{n,k},$$
where **K** is the set of class in the dataset, and $|D_{n,k}|$ = $D_{n,k}$ denotes the total number of samples of class *k* in the MD *n*’s dataset.

As depicted in Figure 6, the overall procedure can be categorized into the following three steps:Step 1: Each MD *n* calculate and send their class distribution vector $\left[D_{n,1},D_{n,2},\ldots ,D_{n,K}\right]$ to the MEC server, where $D_{n,k}$ is the number of the samples with label *k* of MD *n*.Step 2: The MEC server executes the proposed ADM scheme. As step 2-1, using Algorithm 1, dataset size adjustment $D_{n}^{\prime *}$ including $v_{n}^{*}$ is determined. After that, to minimize the $D_{KL}(P_{N}\left|\right|P_{u})$ for making the class distribution of aggregated dataset close to the IID, as step 2-2, class adjustment is conducted. By aggregating such class distribution vector over MDs, the MEC server can calculate the adjustment of data for each class, which is given by
(15)$$reduction_{n,k}=D_{n,k}-reduction_{level},\forall n\in N,\forall k\in K.$$
where $reduction_{n,k}$ represents the reduction applied to each class dataset of MD *n*, and $reduction_{level}$ denotes the target dataset size to be retained across all classes. In this scenario, if (15) yields a value less than 0, it implies that the class size is already below the target dataset size, $reduction_{level}$. Thus, $reduction_{n,k}$ should be set to 0. The process continues until the condition $\sum_{k=1}^{K}D_{n,k}-reduction_{n,k}=v_{n}^{*}\cdot D_{n}$ is satisfied. During this process, $reduction_{level}$ is gradually decreased by subtracting a constant Δ iteratively. Finally, the optimal class distribution vector is updated according to the equation:
(16)$$D_{n,k}^{*}=\left\lceil D_{n,k}-reduction_{n,k}\right\rceil ,\forall n\in N,\forall k\in K.$$Step 3: Finally, as step 3, updated class distribution vector $\left[D_{n,1}^{*},D_{n,2}^{*},\cdots ,D_{n,K}^{*}\right]$ is delivered to the MDs.

The class adjustment algorithm is summarized in Algorithm 2. Here, it is obvious that the time complexity is O(NK), which is simple and easily deployable in the real world with the limited number of MDs and classes, where *N* is the number of MDs and *K* is the number of unique labels.
**Algorithm 2** ADM scheme—class adjustment**Input:** [Dn,1,Dn,2,⋯,Dn,K], vn***Output:** Optimal Dn,k*
 1: **for** each MD n∈N **do**
 2:  $D_{max}=\max\left\{D_{n,1},D_{n,2},\cdots ,D_{n,K}\right\}$
 3:  $reduction_{level}=D_{max}-\Delta$
 4:  **while** $\sum_{k=1}^{K}D_{n,k}-reduction_{n,k}\neq v_{n}^{*}\cdot D_{n}$ **do**
 5:   $reduction_{n,k}=\left\{\begin{array}{ll} D_{n,k}-reduction_{level} & ,\;\mathrm{if}\;D_{n,k}-reduction_{level}\geq 0\\ 0 & ,\;\mathrm{otherwise} \end{array}\right.,k\in K$
 6:   $reduction_{level}=reduction_{level}-\Delta$
 7:  **end while**
 8:  **for** k∈K **do**
 9:   $D_{n,k}^{*}=\left\lceil D_{n,k}-reduction_{n,k}\right\rceil$
 10:   **end for**
 11:  **end for**


### 4.3. Discussion on Implementation of the Proposed ADM on Virtualization Platform: Kubernetes

For practical considerations, this subsection provides a discussion on the implementation of the proposed ADM on Docker-based Kubernetes platforms. As in our previous study [35], the proposed ADM can easily be implemented on a Kubernetes platform consisting of a master and several nodes.

Figure 7 illustrates the implementation architecture of the proposed ADM on the Kubernetes platform. In this example scenario, as depicted in Figure 7, we consider that the architecture comprises one master and two nodes. Specifically, the master is responsible for managing the two nodes using Kubernetes and aggregating the global model for FL. The nodes, on the other hand, are responsible for running pods. Within each pod, there is a container that contains the FL framework received from the master. Consequently, the pod performs local training for the FL by executing the FL framework within the container. To incorporate the ADM scheme into this platform, we also developed the necessary signaling messages for data adjustment (https://github.com/juneseokBang/FL_K8S, accessed on 1 March 2024).

Furthermore, we also implemented a GUI platform that allows users to select FL parameters and compare the trained results using graphical representations (https://github.com/juneseokBang/FL_GUI, accessed on 1 March 2024). Figure 8 illustrates the workflow of the GUI platform. When a user selects a parameter on the web interface, the value is transmitted to the FL server (as shown in Figure 9a,b). The server performs FL using the received parameter value. Once the training is completed, the web interface displays the learning results graphically (as depicted in Figure 9c). Additionally, the platform retains a history of past training results, enabling easy graph comparisons. Although FedAvg is now implemented as a basic algorithm, we can add new parameters or new algorithms to the platform, ensuring its scalability and adaptability to evolving research requirements. This capability empowers researchers to explore a wider range of experimental configurations and evaluate their impact on model performance easily.

## 5. Performance Evaluation

In this section, we present simulation results to validate the effectiveness of the proposed ADM scheme compared to two benchmarks. Benchmark 1 (B1) represents FedAvg applied without any dataset management method, utilizing all training data [1]. Benchmark 2 (B2) represents FedAvg with heuristic dataset management (HDM) [36]. HDM employs a strategy to reduce the size of the training dataset by 20% every 20 rounds without accounting for unbalanced class distribution. We evaluated ADM by training popular CNN models on two datasets. (1) MNIST, a dataset that has 60 K 28 × 28 training images of 10 classes; (2) CIFAR-10, a dataset that contains 50 K 32 × 32 colored images of 10 classes. In the simulated environment, we assume that the computational capacity $f_{i}$ of each MD is 3 GHz, which is fairly assigned to all MDs. Other simulation parameters and hyperparameters are listed in Table 3.

### 5.1. Numerical Analysis—Dataset Size Adjustment

In this subsection, we demonstrate how ADM adjusts dataset size considering **Prob.3** (12a). We assume all MDs have 2500 local datasets and IID data. As shown in Figure 10a,b, $v_{n}$ decreases when the objective function converges (near round 40). In Figure 10a, the objective function converges to zero as the round passes due to discounting factor σ in (Equation 3). Near round 40, $v_{n}$ drops the dataset to 65% because the influence of the dataset on the accuracy gain significantly diminishes. Consequently, not all datasets are required to update the global model.

### 5.2. Simulation Analysis—Dataset Size Adjustment: IID Case

We assume the initial number of training data samples for each MD is 2500 for CIFAR-10 and 3000 for MNIST (i.e., $|D_{n}|$ = 2500 and 3000, respectively), and N is 20, with the data being IID. Table 4 shows the accuracy of ADM and two benchmarks. Data management methods (HDM and ADM) ensure accuracy while reducing the training data, highlighting a significant diminishing effect of dataset size as the rounds progress. However, when the number of MDs is large, and the dataset size is small, the performance of HDM significantly decreases. For CIFAR10, the accuracy of HDM is 50.54%, which is a 6.19% decrease compared to FedAvg. In contrast, ADM achieves an accuracy of 53.73%, demonstrating robustness in scenarios with small datasets. The reduction in HDM’s performance is attributed to its lack of consideration for dataset size, whereas ADM adaptively manages the dataset by taking into account the dataset size of each MD.

As shown in Table 5, when training with all datasets (i.e., B1) for MNIST, it takes about 40.19 s per round, whereas ADM reduces this to approximately 26.88 s per round, representing a 33.1% reduction. Similarly, for CIFAR-10, the time decreases from 35.61 s to 26.71 s, marking a 24.99% reduction. This significantly accelerates the training process. Although HDM reduces the training time more than ADM does, ADM achieves an accuracy comparable to FedAvg (B1), whereas HDM does not guarantee this accuracy. Figure 11 illustrates how ADM adjusts data considering the dataset size. We conducted a total of 100 rounds and averaged every 10 rounds to represent the mean value of vn in each round. If each MD has 1000 data samples, the data reduction rate is not significantly reduced as rounds progress, whereas, with 3000 data samples, the reduction rate drops to 60%. This indicates that the $v_{n}$ is influenced by the dataset size, as outlined in (12a). For $D_{n}$ = 3000 and $D_{n}$ = 2000, volatility increases notably in the 30 s and 40 s rounds, respectively. This suggests that these points are where significant adjustments to $v_{n}$ occur. Therefore, ADM maintains the performance of the local model even with smaller datasets.

### 5.3. Simulation Analysis—Class Adjustment: Non-IID Case

We use β to denote the non-IID level. If β = 0, it indicates that data on each MD uniformly belong to labels. Otherwise, if β = 0.8, it means that 80% of the data belongs to one label, and the remaining 20% of the data belongs to other labels. As shown in Table 4, although HDM guarantees accuracy in non-IID settings when each MD has a large dataset, accuracy significantly decreases with smaller datasets. In contrast, ADM considers class distribution at all dataset sizes, reducing larger classes to achieve near-IID conditions and ensuring accuracy. Figure 12a,b demonstrate how ADM adjusts data based on varying data sizes and class distribution across MDs, respectively. MDs 1–10 have 1000 datasets each, while MDs 11–20 have 3000 datasets each. All MDs have non-IID data with β = 0.8. As depicted in Figure 12a, MDs with 1000 datasets reduced to 67% of the existing dataset (represented by the blue bars), while MDs with 3000 datasets decreased even further to 48% of the existing dataset (represented by the green bars). These reductions occurred because ADM adjusts dataset sizes based on the individual sizes of each MD’s dataset. Figure 12b illustrates the number of data for each label within the MD with 1000 datasets. Instead of reducing the dataset size for labels with smaller datasets, our approach balances the dataset size with the IID by reducing the data for labels with larger datasets. In conclusion, our proposed ADM effectively adjusts datasets in both IID and non-IID scenarios, alleviating the burden of local training on MDs.

## 6. Conclusions

In this study, we explored the impact of dataset size on accuracy gain as the communication round evolves. Based on this insight, to alleviate the burden on local training on MDs for FL, we introduced a novel ADM scheme. We incorporated a discount factor to quantify the diminished influence of dataset size on accuracy gain across communication rounds. Specifically, the proposed ADM scheme includes both dataset size adjustment and class adjustment to optimize how dataset reduction is applied across different classes. This optimization takes into account both the discount factor and the KLD. Our experimental results show that our ADM scheme significantly reduces the training burden on MDs while maintaining acceptable training accuracy. In our future work, given that FL remains susceptible to privacy attacks wherein adversaries could potentially retrieve raw data by inspecting local model updates, it is imperative to integrate robust privacy protection mechanisms into our framework (i.e., differential privacy, etc.). Additionally, malicious FL participants pose a significant threat, as they may attempt to inject poisoned or noisy models into an FL server. To mitigate this risk, deploying the proposed scheme on a blockchain platform could be advantageous. Leveraging the inherent integrity of blockchain records ensures that any malicious activities by FL participants can be traced back, as the records remain untampered.

## Figures and Tables

**Figure 1 sensors-24-02579-f001:**
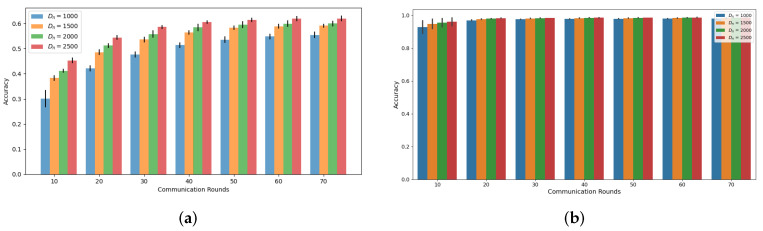
Accuracy based on communications rounds and $D_{n}$ on IID setting (**a**) CIFAR-10 (**b**) MNIST.

**Figure 2 sensors-24-02579-f002:**
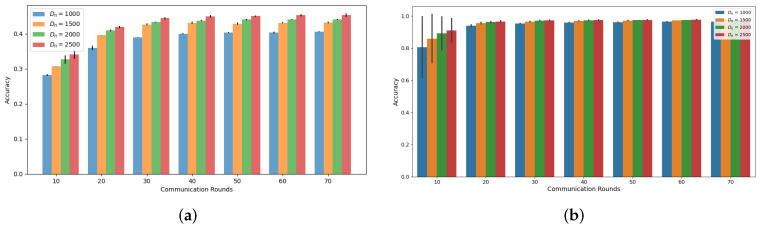
Accuracy based on communications rounds and $D_{n}$ on Non-IID setting (**a**) CIFAR-10 (**b**) MNIST.

**Figure 3 sensors-24-02579-f003:**
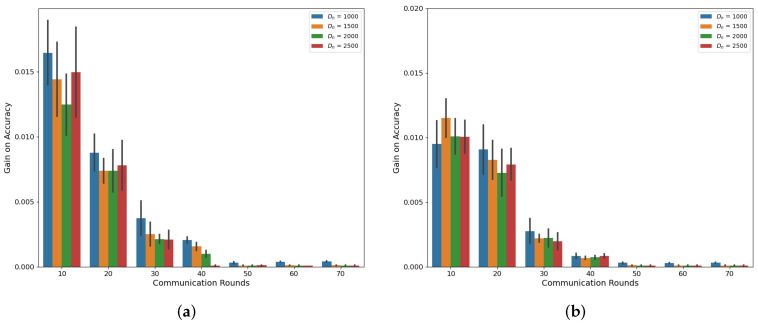
Accuracy gain based on communications rounds and $D_{n}$ on CIFAR-10 (**a**) IID (**b**) Non-IID.

**Figure 4 sensors-24-02579-f004:**
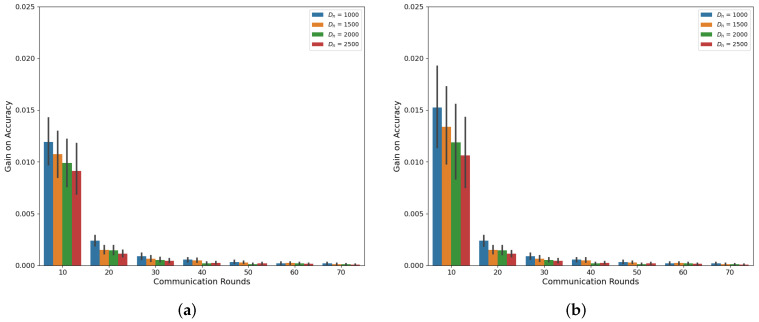
Accuracy gain based on communications rounds and $D_{n}$ on MNIST (**a**) IID (**b**) Non-IID.

**Figure 5 sensors-24-02579-f005:**
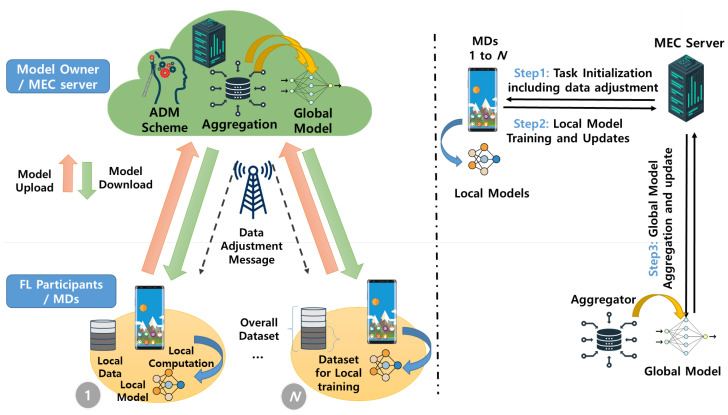
Proposed system architecture.

**Figure 6 sensors-24-02579-f006:**
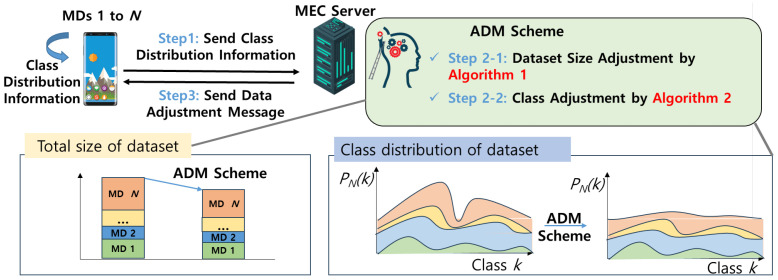
The overall process of the ADM scheme.

**Figure 7 sensors-24-02579-f007:**
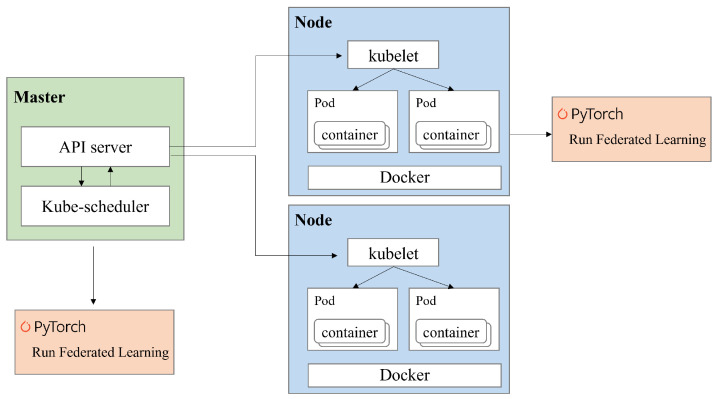
Implementation of the proposed ADM on a virtualization platform.

**Figure 8 sensors-24-02579-f008:**
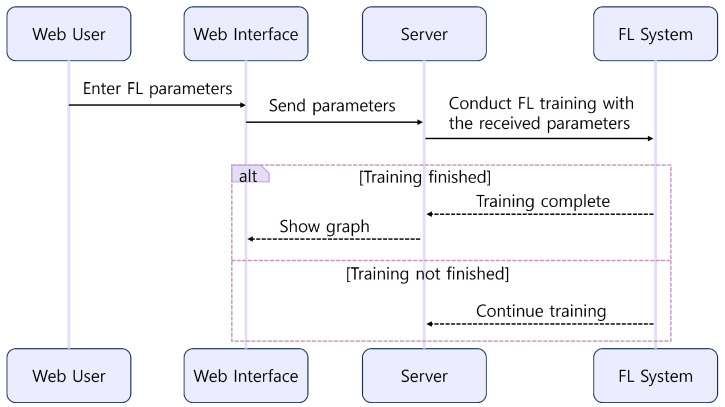
GUI Platform Workflow.

**Figure 9 sensors-24-02579-f009:**
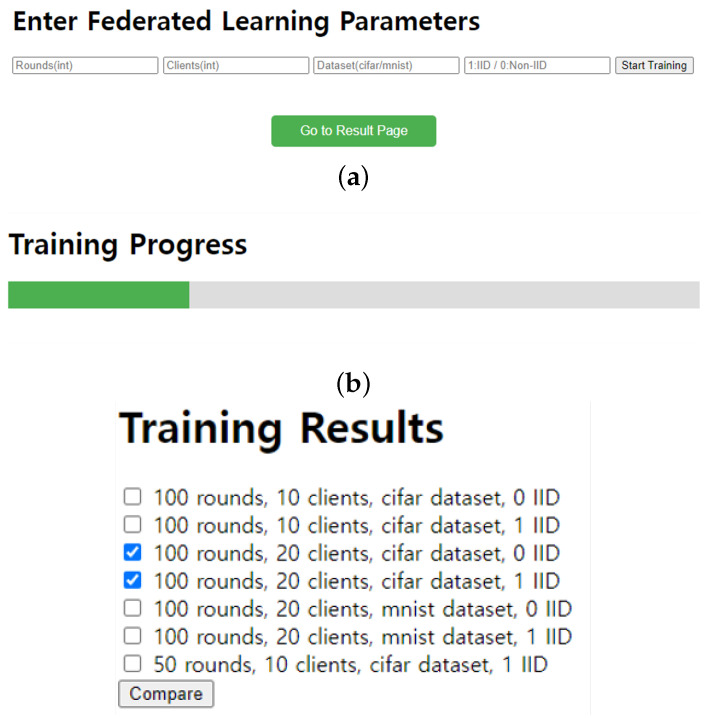
GUI platform. (**a**) entry of FL hyperparameters (**b**) training progress (**c**) training result comparison.

**Figure 10 sensors-24-02579-f010:**
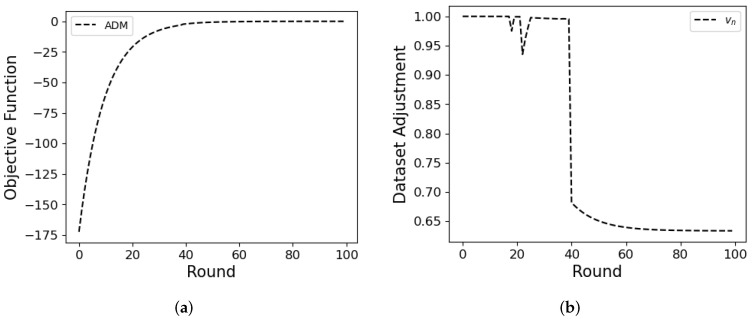
Objective function (**a**) and the average dataset adjustment $v_{n}$ (**b**) over rounds.

**Figure 11 sensors-24-02579-f011:**
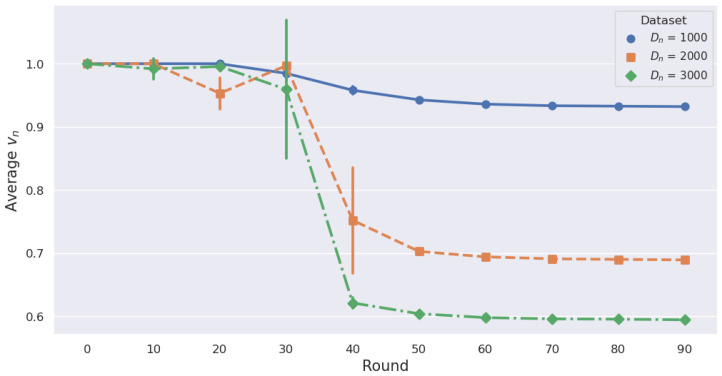
Average Data Reduction Rate per 10 Rounds for Different Datasets with ADM.

**Figure 12 sensors-24-02579-f012:**
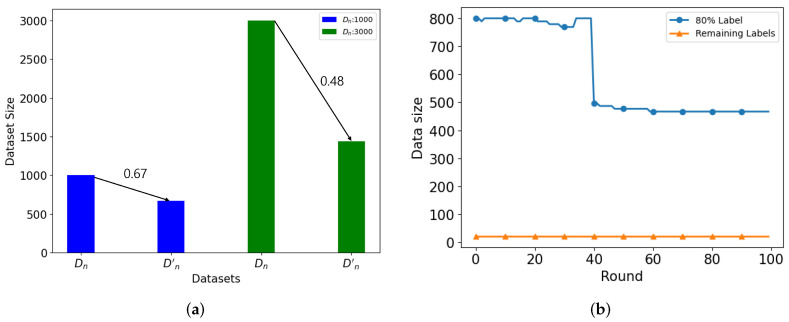
(**a**) Dataset Reduction based on MD’s Dataset Size, (**b**) Dataset Reduction based on Class Distribution.

**Table 1 sensors-24-02579-t001:** Summary of related works.

Related Works	Topic	Key Contributions
[6]	Learning Efficiency	Propose dataset and computation management strategy
[7]	Data Distribution Management	Augment the minority class and downsample the majority class
[9]	Learning Efficiency	Propose batch size selection and radio resource allocation algorithm
[26]	Learning Efficiency	Propose an adaptive batch size and learning rate selection algorithm
[27]	Learning Efficiency	Choose the data contributing to the improvement of the model
[28]	Data Distribution Management	Propose data augmentation strategy using GAN to promote IID data
[29]	Data Distribution Management	Swap models among the MDs to alleviate data distribution
[30]	Data Distribution Management	Select clients with a low degree of non-IID data using weight divergence
[31]	Data Distribution Management	Select clients relevant to the application task employing DRL
ADM	Learning Efficiency Data Distribution Management	Balance accuracy and latency by adjusting dataset size Propose a method for diminishing the effect of dataset size over rounds Adjust class distribution on non-IID data

**Table 2 sensors-24-02579-t002:** Summary of Major Symbols.

Symbol	Definition
**N**	Set of MDs
*N*	Total number of MDs
**D** _ **n** _	Dataset of MD *n*
$D_{n}$	Total number of samples in MD *n*’s dataset
$D_{n}^{\prime }$	Adjusted number of samples in MD *n*’s dataset
σ	Discounting factor
*A*	Accuracy estimation model
$c_{n}$	Number of CPU cycle of MD *n* required to process one samples
$f_{n}$	CPU frequency of MD *n*
$L_{n}^{c}$	Local model computation latency
$I_{n}$	Number of training local model iteration
*B*	Available bandwidth at MEC server
$R_{n}$	Transmission rate of MD *n*
$W_{n}$	Size of the local model parameters of MD *n*
*L*	End-to-end service latency of FL
$v_{n}$	Dataset adjustment vector
$D_{KL}$	Kullback–Leibler divergence distance
$v_{n,k}$	Class adjustment vector

**Table 3 sensors-24-02579-t003:** Simulation Parameters.

Parameter	Value
Number of CPU cycles ($c_{n}$)	30 cycles/sample
Computation capacity ($f_{n}$)	3 GHz
Noise power ($N_{0}$)	−114 dBm
Size of local model ($W_{n}$)	100 Kbits
Gamma ($\gamma_{d}$)	0.4
Number of MDs (*N*)	20
Discounting factor (σ)	0.9 × 10^−8^

**Table 4 sensors-24-02579-t004:** Test Accuracy.

Method	Data Distribution	MNIST (N = 20, $D_{n}$ = 3000)	CIFAR-10 (N = 20, $D_{n}$ = 2500)	MNIST (N = 50, $D_{n}$ = 1250)	CIFAR-10 (N = 50, $D_{n}$ = 1000)
FedAvg (B1)	IID	99.32%	54.80%	98.83%	53.88%
	Non-IID (β = 0.8)	98.98%	49.63%	98.48%	49.54%
HDM (B2)	IID	99.20%	54.20%	98.54%	50.54%
	Non-IID (β = 0.8)	98.86%	49.59%	98.04%	48.53%
ADM	IID	99.25%	54.65%	98.68%	53.73%
	Non-IID (β = 0.8)	98.92%	49.54%	98.35%	49.25%

**Table 5 sensors-24-02579-t005:** Average Training Time when N = 50 on IID case.

Second/Round (Average)	MNIST ($D_{n}$ = 1250)	CIFAR-10 ($D_{n}$ = 1000)
FedAvg (B1)	40.19 s	35.61 s
HDM (B2)	24.34 s	22.14 s
ADM	26.88 s	26.71 s

## Data Availability

Not applicable.

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
