# Peer review of "Adaptive Dataset Management Scheme for Lightweight Federated Learning in Mobile Edge Computing"

_sensors, 2024, doi:10.3390/s24082579_

Round 1
Reviewer 1 Report
Comments and Suggestions for Authors
The authors propose an adaptive dataset management (ADM) scheme to reduce the burden of local training on MDs. The proposal is interesting. However, the experiments must be improved to support some conclusions. Some comments are below.
1) Related Work: The subsection "Computation (CPU/GPU) and radio resource management" is not directly related work. It can be merged for contextualization in the introduction, for example.
2) Related Work: The subsection "Dataset management" must be improved. The contribution of the paper is not evident. The authors must describe the contributions and differences to the related work. A comparative table will be interesting.
3) Figure 3: The confidence interval is not good. The authors must explain it.
4) Only a few experiments do not support the affirmative: "Based on these findings, we can conclude that in order to alleviate the training burden on MDs, it is advisable to more aggressively reduce the dataset size as the communication rounds evolve. " The authors should perform variations such as other datasets, among others.
5) The authors should use a formal notation (e.g. PBMN) in Figures 4 and 5.
6) The training was performed on 10 and 20 clients (mainly). Why? How would the proposal perform with more clients? The variation is important to evaluate scalability and adaptative features.
7) Figure 10 needs a confidence interval to evidence the variation (Figure 10. Average Data Reduction Rate per 10 Rounds for Different Datasets with ADM).
Comments on the Quality of English Language
Minor editing of English language required.
Reviewer 2 Report
Comments and Suggestions for Authors
This paper proposes an Adaptive Dataset Management (ADM) scheme for federated learning (FL) on mobile devices (MDs) to minimize accuracy loss while reducing the training load. Some potential areas for development are listed below:
1. Contribution consideration: The paper's structure and steps are explained in the introduction section and the contributions part : "The detailed contributions of this article are as follows:", , but it does not make any genuinely original contributions; instead, it focuses more on the explanation of steps.
2. Research background: Although the related work section is thorough, it could be made better by mentioning the following paper: https://onlinelibrary.wiley.com/doi/abs/10.1002/dac.5267
3. Security and privacy considerations: Be aware of the risks associated with federated learning. Briefly explain how the ADM scheme addresses these problems or make recommendations for any potential integrated mitigation strategies.
Reviewer 3 Report
Comments and Suggestions for Authors
The authors provided a solid evaluation, where both empirical analysis and theorems are provided to corroborate the authors' findings (two proposed algorithms). This implementation was tested within a simulated environment.
Despite the authors using a dataset for a competing algorithm, FedAvg, it is not clear whether this algorithm was the actual "conventional method" referenced in Benchmark 1/2 or not. Clarifying this is straightforward, as it would only involve adding an explicit reference and the name for the algorithm associated with B1 and B2 rather than providing an uninformative reference. While providing the experiments in Table 4 and narrating it in the text (Sect. 5.2), the authors should also remember why B2 was not reported; furthermore, it would help with clarity if the dimensions associated with the rows/columns of the tables are kept consistent throughout the paper (e.g., the algorithms always in the header and the dataset and hyperparameters on the rows, or vice-versa).
Also, as both B1 and the proposed ADM seem to consider data reduction properties from Table 4, it puzzles the reviewer why the results from Figure '10 are considered just for ADM and not for B1: similar considerations can also be applied to Figure 11. This will help the reader understand the contribution of the current paper better while providing an all-around comparison that considers not only the properties of the proposed approach but also the referenced state of the art.
Despite the authors sincerely appearing to have provided a solid and all-around paper, better clarity in the experiment section will corroborate and substantiate the overall effort with more explicit comparisons across the algorithms.
Round 2
Reviewer 1 Report
Comments and Suggestions for Authors
The authors answered all my questions and addressed the issues. The paper is acceptable.
Reviewer 2 Report
Comments and Suggestions for Authors
The revised version is good now, and in my opinion it can be published.
Reviewer 3 Report
Comments and Suggestions for Authors
The authors clarified the previous points of the discussion, such as the competing algorithms while clarifying some omissions (i.e., as the prior competitor #2 did not require training time for dimensionality reduction, this was omitted from the rest of the paper).
Formal proofs corroborate the correctness of the proposed approach, which is still evaluated experimentally/in practice, albeit in a simulated environment (but this is rather customary in the field if you don't have a real system where to test the algorithm).
The authors addressed all my questions and can be accepted in the present form.